# Therapeutic Potential of a Senolytic Approach in a Murine Model of Chronic GVHD

**DOI:** 10.3390/biology12050647

**Published:** 2023-04-25

**Authors:** Deepika Raman, Charlotte Chêne, Carole Nicco, Mohamed Jeljeli, Jie Qing Eu, Marie-Véronique Clément, Frédéric Batteux, Shazib Pervaiz

**Affiliations:** 1Department of Physiology, Yong Loo Lin School of Medicine, National University of Singapore, Singapore 117597, Singapore; deepika_raman@eddc.a-star.edu.sg (D.R.);; 2Département 3I, Infection, Immunité et Inflammation, Institut Cochin, INSERM U1016, Université de Paris, 75014 Paris, France; 3Université de Paris, Faculté de Médecine, AP-HP-Centre Université de Paris, Hôpital Cochin, Service d’Immunologie Biologique, 75014 Paris, France; 4Cancer Science Institute, National University of Singapore, Singapore 117597, Singapore; 5Department of Biochemistry, Yong Loo Lin School of Medicine, National University of Singapore, Singapore 117597, Singapore; bchmvc@nus.edu.sg; 6NUS Medicine Healthy Longevity Program, National University of Singapore, Singapore 117597, Singapore; 7Integrated Science and Engineering Program, NUS Graduate School, National University of Singapore, Singapore 117597, Singapore; 8NUS Centre for Cancer Research (N2CR), Yong Loo Lin School of Medicine, National University of Singapore, Singapore 117597, Singapore; 9National University Cancer Institute, National University Health System, Singapore 117597, Singapore

**Keywords:** graft vs. host disease, quercetin, dasatinib, senescence

## Abstract

**Simple Summary:**

Graft-versus-host disease is a potentially life-threatening complication after bone marrow transplantation from an unrelated donor. The multi-organ damage is triggered by the donor cells that attack the host tissue. Patients manifest signs and symptoms in the skin, liver, lungs and other body organs. Treatment is usually by suppressing the immune system to limit foreign immune cells attacking the host tissues/organs. In this pilot study, we hypothesized that populations of cells termed senescent cells that produce inflammatory proteins might contribute to the disease pathology. Therefore, using a mouse model of chronic graft-versus-host disease, we tested a drug combination that has been shown to target and kill such populations. These types of agents are referred to as senolytics. We show promising therapeutic efficacy of this combination approach in the mouse model of disease, which could have implications for human disease.

**Abstract:**

Graft-versus-host disease (GVHD) is a life-threatening systemic complication of allogeneic hematopoietic stem cell transplantation (HSCT) characterized by dysregulation of T and B cell activation and function, scleroderma-like features, and multi-organ pathology. The treatment of cGVHD is limited to the management of symptoms and long-term use of immunosuppressive therapy, which underscores the need for developing novel treatment approaches. Notably, there is a striking similarity between cytokines/chemokines responsible for multi-organ damage in cGVHD and pro-inflammatory factors, immune modulators, and growth factors secreted by senescent cells upon the acquisition of senescence-associated secretory phenotype (SASP). In this pilot study, we questioned the involvement of senescent cell-derived factors in the pathogenesis of cGVHD triggered upon allogeneic transplantation in an irradiated host. Using a murine model that recapitulates sclerodermatous cGVHD, we investigated the therapeutic efficacy of a senolytic combination of dasatinib and quercetin (DQ) administered after 10 days of allogeneic transplantation and given every 7 days for 35 days. Treatment with DQ resulted in a significant improvement in several physical and tissue-specific features, such as alopecia and earlobe thickness, associated with cGVHD pathogenesis in allograft recipients. DQ also mitigated cGVHD-associated changes in the peripheral T cell pool and serum levels of SASP-like cytokines, such as IL-4, IL-6 and IL-8Rα. Our results support the involvement of senescent cells in the pathogenesis of cGVHD and provide a rationale for the use of DQ, a clinically approved senolytic approach, as a potential therapeutic strategy.

## 1. Introduction

Graft-versus-host disease is a complication of allogeneic transplantation and affects approximately half of recipients. Chronic GVHD (cGVHD) usually starts within the first 100 days after transplantation and is characterized by immune-mediated inflammatory damage, manifested as a multi-organ disease of major target organs such as the skin, liver, eyes, and oral cavity, although other organs such as the gut and lungs may also be involved [1,2]. As such, the use of immunosuppressive drugs is the primary strategy to prevent cGVHD, which is fraught with increased susceptibility to infections and relapse, the major causes of mortality following bone marrow transplantation. The cellular mediators of organ damage include various CD4^+^ T cell subsets such as Th1, Th2, Th17 and CD8^+^ T cells [3]. Notably, the presence of circulating anti-nuclear and other autoantibodies also points to an abnormal activation of B cell-mediated immune responses [4]. This is corroborated by clinical evidence of reduced naïve and memory B cells, an abnormal increase in B cell activation factor (BAFF), and the clinical efficacy of rituximab (anti-CD20) in steroid-refractory cGVHD [4,5,6]. Furthermore, a slew of cytokines and chemokines produced by alloreactive T cells, such as TNF-α and IFN-γ (Th1 cells), IL-17, IL-21, and IL-22 (Th17 cells), IL-4 and IL-13 (Th2 cells), and IL-17A, CXCL9, and CXCL10 (CD8^+^ cells), collectively contribute to the massive inflammatory response and organ damage in cGVHD [3].

While the systemic manifestations of the disease are associated with immune dysregulation, there is also emerging evidence to implicate senescent cells in the etiology and/or pathophysiology of the disease [7,8]. Senescence is a stress-induced cellular response resulting in irreversible cell cycle arrest in cells that continue to be metabolically active, as evidence by amplified mTOR (mammalian target of rapamycin) activity [9]. Accumulation of a senescent cell pool is associated with organismal aging, and recent evidence indicates a role in promoting as well as regulating the process of carcinogenesis [10,11,12]. The biological effects elicited by senescent cells are a function of senescence-associated secretory factors that are released upon the acquisition of SASP [13]. Notably, amongst the factors released by senescent cells are pro-inflammatory cytokines, such as IL-6, IL-8, and CXCL9, which are active drivers of cGVHD [7]. A cause-and-effect relationship between SASP factors and cGVHD is corroborated by the significant improvement and/or prevention of tissue damage by senolytic drugs, such as the Bcl-2 inhibitor navitoclax (ABT263) [14].

In the current study, we evaluated the therapeutic potential of a senolytic combination approach as a proof of concept against the systemic manifestations of cGVHD. Using a murine model of allogeneic bone marrow (BM) transplantation in which BM cells from MHC-mismatched donor B10.D2 mice were transplanted into sublethally irradiated BALB/c mice, we evaluated the effect of simultaneous administration of dasatinib and quercetin (DQ), a combination approach that has shown promise against several senescence-mediated pathologies [15,16,17,18,19]. We report association of cGVHD-related changes in the immune cell repertoire and organ pathology, with an increase in expression of IL-8 and IL-6. Notably, combination therapy with DQ resulted in a remarkable rescue of skin manifestations as well as a significant decrease in IL-8, IL-6, and IL-4, particularly in the liver and skin.

## 2. Methods

### 2.1. Establishing a Murine Model of cGVHD

Eight-week-old female BALB/c (H2^d^) mice (Janvier Laboratory, Le Genest-Saint-Isle, France) and male B10.D2 (H2^d^) mice kindly offered by Colette Kanellopoulos-Langevin (CDTA, CNRS, Orléans, France) were used in all experiments. Mice were treated with humane care in compliance with institutional ethical guidelines (Inserm and Université Paris Descartes—CEEA34 Ethics Committee). All mice were housed in ventilated cages with sterile food and water ad libitum. Subsequent transplantation of splenocytes and bone marrow cells into BALB/c mice (H-2^d^; Janvier Laboratory) by grafting cells from male B10.D2 mice (H-2^d^; Janvier Laboratory) led to the development of GVHD, as previously described [20,21,22]. Briefly, host mice were lethally irradiated with 750 cGy from a Gammacell (^137^Cs) source. After 3 h, host mice were injected with donor spleen cells (2 × 10^6^ cells) and bone marrow cells (1 × 10^6^ cells) previously treated with a hypotonic solution of potassium acetate for red blood cell lysis and suspended in RPMI 1640. A brief schematic of the model is presented in Figure 1A. A control group of BALB/c recipient mice received syngeneic BALB/c spleen and BM cells. Mice with allogeneic bone marrow transplantation were left untreated for 10 days (average time for the development of murine GVHD) before being randomized and treated every 7 days for 35 days with either DQ (5 mg/kg dasatinib and 50 mg/kg quercetin) [15,19,23] by oral gavage or vehicle (water) alone (7 mice/group). Notably, treatment of all mice with or without clinical development of murine cGVHD was started 10 days following the transplantation. Mice in the irradiation control group died within 3 days. Animals were routinely weighed, and clinical symptoms were recorded up to 3 times a week for each group. Mice were humanely sacrificed on day 37 and tissues and sera were collected for qRT-PCR, histological analyses, ELISA and immune cell profiling.

### 2.2. Disease Severity Index

To determine the incidence and severity of disease, we assigned a score to each mouse using the following criteria: 0: no external sign; 1: alopecia; 1: piloerection and/or hunched posture; 1: vasculitis (one or more purpuric lesions on the ears or tail) or eyelid sclerosis (blepharophimosis); and 1: diarrhea. The severity score was calculated as the sum of these values, ranging from 0 (unaffected) to a maximum of 4. Disease severity scores were routinely recorded and plotted for the DQ-treated and untreated mice within the allogeneic transplant recipients.

### 2.3. Earlobe Thickness

Ear thickness was measured up to three times a week with the aid of an electronic digital caliper and expressed in millimeters.

### 2.4. Flow Cytometric Analysis of Spleen Cell Subsets

Cell suspensions from spleens were prepared after hypotonic lysis of erythrocytes with potassium acetate solution. Cells were incubated with the appropriate labeled antibodies (Abs) at 4 °C for 30 min in phosphate-buffered saline (PBS) with 2% normal fetal calf serum. Flow cytometry was performed using a FACS Canto II flow cytometer (BD Biosciences, Franklin Lakes, NJ, USA), according to standard techniques. The monoclonal Abs used in this study were anti-B220-PerCp-Cy5.5, anti-CD4-PE-Dazzle, anti-CD8-APC-Cy7, anti-CD62L-PE-Cy5, and anti-CD44-APC (BD Biosciences). Data were analyzed with FlowJo software (Tree Star, Ashland, OR, USA).

### 2.5. Histopathological and Immunohistochemistry Analyses

Fixed tissue (liver, ear, and skin) biopsies were embedded in paraffin. The tissue sections were stained with hematoxylin and eosin (H&E). Fixed earlobes and skin sections were further stained with picrosirius red. Slides were examined by standard bright-field microscopy (Olympus BX53, Tokyo, Japan). Liver sections for CD45 (Santa Cruz Biotechnology Inc., Dallas, TX, USA; #sc1178) were subjected to citrate antigen retrieval. Immunohistochemistry was performed using the BOND-MAX Automated Immunohistochemistry Biosystem (Leica Biosystems, Wetzlar, Germany). Briefly, tissues were deparaffinized and subjected to antigen retrieval and peroxidase blocking. The biopsies were next incubated with primary Ab in Ab diluent with background-reducing reagent for 8 min, followed by subsequent incubation with polymer for 8 min and DAB for 10 min prior to hematoxylin staining for 10 min. For all other Abs (Col1A1, MPO, p16), the biopsies were subjected to Tris-EDTA antigen retrieval protocol and stained for the primary Ab or control isotype overnight according to standard manufacturer’s instructions using a Dako kit (EnVision™ + Dual Link System HRP DAB+). Ear and skin tissue sections were stained for Col1A1 (Sigma-Aldrich, St Louis, MO, USA; #SAB5700733) and skin and liver biopsies were stained for MPO (Santa Cruz Biotechnology Inc., Dallas, TX, USA; # sc-390109). Finally, liver biopsies were analyzed for p16 (Thermo Fisher Scientific, Waltham, MA, USA; #MA5-17142) expression.

### 2.6. Multiplex Protein ELISA

Samples were subjected to a 14-plex ELISA (Customized ProcartaPlex^TM^ immunoassay, Thermo Fisher Scientific, Waltham, MA, USA) according to the standard manufacturer’s protocol and multiplex protein quantitation was performed using the Luminex instrument platform. Samples were further subjected to IL8Rα (CXCR1) ELISA (MyBiosource, San Diego, CA, USA), and the standard manufacturer’s protocol was followed. Results were quantitated using a Tecan spectrophotometer (O.D. 450 nm).

### 2.7. cDNA Synthesis and qPCR

Tissues were harvested in Trizol reagent and RNA extraction was performed using a bead-rotor homogenizer, followed by chloroform–isopropanol–ethanol precipitation. Following this, cDNA synthesis was performed using either a Maxima first-strand cDNA synthesis kit or OneScript Plus cDNA synthesis kit, according to the manufacturer’s protocol. Samples were then run using SYBR Select master mix on a 384-well plate format on LightCycler480-II (Roche, Basel, Switzerland) platform.

### 2.8. Statistical Analysis

All quantitative data are expressed as means ± SD. Data were compared using ANOVA unless otherwise specified. Student’s unpaired *t*-test was used to compare two groups. All analyses were carried out using the GraphPad Prism statistical software package (GraphPad Software, Inc., San Diego, CA, USA). Significance was set at *p* < 0.05.

## 3. Results

### 3.1. DQ Alleviates Adverse Effects of Allogeneic Graft on the General Well-Being of the Recipient

Following the establishment of the allogeneic transplant model, general physical well-being of the animals (appearance and weight) and disease severity index (described in Methods) were routinely monitored. Firstly, treatment with DQ had significantly mitigated the effect of allogeneic transplantation on the overall appearance and body weight of the mice at 35 days following grafting (Figure 1B,C). No significant difference in appearance or body weight was observed between mice that received syngeneic grafts and those that were treated with DQ after allogeneic grafts. Secondly, assessment of the disease severity index revealed a significantly lower overall score in DQ-treated animals from day 24 to day 35 compared to the untreated allogeneic group (Figure 1D and Table 1).

### 3.2. DQ Mitigates Allograft Associated Skin Fibrosis

We next assessed DQ’s ability to mitigate established signs of murine cGVHD, including ear/skin thickness, an index of skin fibrosis and one of the classical signs associated with cGVHD in mice. Results clearly showed an increase in ear thickness in allogeneic graft recipients compared to the syngeneic controls, which was notably reduced upon treatment with DQ (Figure 2A). Corroborating that, H&E staining of tissue sections from the ears of mice in the respective groups indicated enhanced fibrosis in allogeneic recipients (cGVHD) compared to the syngeneic group, which was partially reduced upon treatment with DQ (Figure 2B–D and Appendix A). The latter was further validated by picrosirius red staining and collagen (Col1A1) expression, which confirmed the inhibitory effect of DQ on increased collagen content upon allogeneic grafting (Figure 2C,D and Appendix A). A similar effect of DQ was observed in skin sections of allogeneic graft recipients (Figure 2E–G and Appendix A). Skin sections were further examined for MPO^+^ neutrophil infiltration. Interestingly, allograft recipient mice displayed lower MPO^+^ staining as compared to their syngeneic counterparts, suggesting the clearance of exhausted neutrophils at the late stage of cGVHD (Figure 2H and Appendix A). We were unable to discern a significant difference in α-SMA in ear and skin sections between the allogenic and syngeneic graft recipients; however, a significant reduction was observed in the DQ-treated allogeneic graft recipients (Appendix A). Moreover, the marginal increase in p16 expression in skin sections of allogenic recipients was reduced upon treatment with DQ (Figure 2I and Appendix A).

### 3.3. DQ Significantly Inhibits Allograft Associated Increase in Circulating Memory T Cells without Significantly Affecting the Decrease in B Cell Subsets

Previous studies have reported reduced B cell precursors, diminished population of peripheral blood CD27+ B cells and lower IgA and IgG2 immunoglobulin levels in cGVHD patients as well as lower rates of B cell reconstitution upon allogeneic HSCT [4,24,25]. In line with previous studies, a significantly lower percentage of B cells (B220^+^) was seen in allograft recipients, which was not significantly affected by DQ treatment (Figure 3A). Since cGVHD has been attributed to the reaction of the recipient to the donor T cells, we next sought to investigate if the clinical improvement observed in allograft recipient mice subjected to DQ treatment correlated with quantitative or qualitative alterations in T cell populations. Flow cytometric analysis indicated that the marked increase in CD44^+^ CD62L^−^ effector memory CD4^+^ cells upon allogeneic transplantation was significantly reduced in animals treated with DQ (Figure 3B). Similarly, while no perceptible change was noted in the total CD8^+^ T cell population, DQ was able to significantly reduce the population of effector memory CD8^+^ cells (Figure 3C). The gating strategy used for FACS analysis for sorting and analyzing lymphocyte subsets is shown in Appendix A.

### 3.4. DQ Mitigates the Increase in IL-4, IL-6 and IL-8Rα upon Allografting

Having observed the effect of DQ on allograft-induced circulating immune cell profile, we next assessed the effect on serum levels of cytokines and chemokines using a multiplex ELISA approach confirmed by specific measurements of the affected cytokines. Results showed an increase in IL-4 and IL-6 and a decrease in IFN-γ and IL-17A in allograft recipients compared to their syngeneic counterparts (Figure 4A). DQ treatment prevented the increases in IL-4 and IL-6 without significantly affecting the decrease in IFN-γ or IL17A (Figure 4A). Single-cytokine ELISA also confirmed the mitigating effect of DQ on IL-8Rα (Figure 4B).

### 3.5. DQ Targets Allograft-Associated Senescent Cell Pool

Intrigued by the effect of DQ on serum levels of cytokines such as IL-4, IL-6 and IL-8Rα, which have been associated with SASP, we next queried if the beneficial effect of DQ was a function of its ability to target a pool of senescent cells induced upon allografting. We analyzed tissue sections obtained from liver, colon, and lungs for mRNA levels of IL-6 and IL-8. Results showed a significant increase in IL-6 or IL-8 in the liver of allografted mice compared to the syngeneic recipients (Figure 5A). While DQ treatment virtually completely blocked the increase in IL-8, a modest effect was also observed on IL-6 (Figure 5B).

To verify the involvement of senescence and the effect of DQ treatment on it, next we looked at the expression of a bona fide senescence-associated protein, p16, in tissue sections obtained from the livers and skin of respective groups. A significant increase in p16^+^ cells was observed in allograft recipient livers, which was strongly inhibited by DQ (Figure 5D). Furthermore, while p16^+^ cells were detected in the skin sections of syngeneic and allogeneic graft recipients, DQ treatment was able to further reduce p16^+^ cells in the allograft recipients, supporting the senolytic activity of DQ (Appendix A). Notably, the recruitment of p16^+^ cells in the liver upon allografting strongly correlated with the appearance of CD45^+^ cells within the liver, which could be significantly reduced by DQ (Figure 5E), thus arguing in favor of a stimulatory role of a senescent pool in recruiting alloreactive CD45^+^ cells in the liver. As DQ efficiently targets senescent cells, the marked reduction in p16, a marker of senescence, seems to suggest purging of p16^+^ cells; however, this warrants further investigation and may be determined by using a p16+ or SA-β-Gal^+^ reporter [1,2] model system to track senescent cell population(s) in cGVHD and the effect of DQ. We further observed significant increases in IL-1β expression in the skin and IL-1α and CDKN2A (senescence marker) in the intestines of allografted mice (Appendix A). Only intestinal CDKN2A expression was significantly reduced in the allograft recipients subjected to DQ treatment (Appendix A).

## 4. Discussion

In this study, we set out to test the therapeutic efficacy of the senolytic combination of dasatinib and quercetin (DQ) in an experimental murine model of cGVHD. Experimental cGVHD was induced in sublethally irradiated BALB/c mice upon transplantation of allogeneic grafts from B10.D2 mice. The typical characteristics of murine cGVHD include skin and ear fibrosis, eyelid sclerosis, alopecia, changes in the immune profile, and the presence of pro-inflammatory cytokines in major organs such as liver, lungs, and gut [26,27]. The phenotypic changes and organ pathology bear striking similarities with clinical cGVHD, an autoimmune-like inflammatory disease that typically affects multiple organs, including skin (75%), oral mucosa (51–63%), liver, eyes and the gastrointestinal (GI) tract (22–51%) [28].

Organ damage and tissue injury such as collagen deposition and fibrosis are brought about by donor lymphocyte populations and involve profibrotic mediators and inflammatory cytokines [29]. Interestingly, inflammation and fibrosis of the liver and GI tract are hallmarks of cGVHD, with macrophage infiltration as an important biomarker [30,31]. Our results provide evidence that DQ treatment, subsequent to the development of cGVHD, resulted in a marked improvement in the physical appearance of the recipient mice as well as reduced fibrotic changes and collagen deposition. Notably, the significantly elevated levels of IL-8Rα mRNA in the livers of allograft recipient mice was drastically reduced upon treatment with DQ. IL-8 (and IL-8 like chemokines in mice) is an integral pro-inflammatory cytokine secreted by macrophages and shown to promote neutrophil recruitment and activation that triggers tissue and organ damage in several pathologies such as COPD, asthma, scleroderma and cystic fibrosis [32,33]. IL-8 is also an important marker of leukocyte migration [34,35]. Furthermore, a significant reduction in CD45^+^ cells was observed in the liver of allografted mice upon DQ treatment, thus indicating an effect on lymphocyte infiltration in the liver. Several studies have associated an increase in memory T cells with the onset or progression of cGVHD; however, its specific role in the underlying etiology of disease has not been demonstrated [36,37,38,39,40]. DQ treatment significantly alleviated the increase in circulatory memory CD4^+^ and CD8^+^ T cells (CD44^high^ CD62L^low^) in allograft recipients. These results are suggestive of a mechanism of action of DQ upstream of the recruitment of alloreactive lymphocytes into the recipient tissues/organs, thereby limiting some of the systemic manifestations of cGVHD.

Interestingly, alopecia and other changes in the skin are classical hallmarks of radiation-induced DNA damage that triggers inflammation as well as promotes the acquisition of senescence and SASP [41]. Together, inflammatory mediators (IL-17 and CCL20) and SASP factors such as IL-1 and IL-6 function in a self-amplifying loop to induce alopecia and dermatitis in irradiated animals [41]. Interestingly, senescence and SASP induction have been closely associated with the pathogenesis of cGVHD [7,42]. Consistent with that, expression of the senescence-associated protein p16 was observed in the liver of allografted mice, which was strikingly diminished upon treatment with DQ. Furthermore, the significant increases in secreted levels of IL-4, IL-6 and IL-8 provide testimony to the presence of a pool of senescent cells upon allografting, as these cytokines are associated with the acquisition of SASP. One might conjecture that the concomitant increase in CD45^+^ cells in the liver upon allograft transplantation could be a function of factors secreted by senescent cells, and the fact that DQ significantly alleviated this increase suggests that the senolytic activity of DQ eliminates cells that tend to provide the stimulus for CD45^+^ cell recruitment in the liver. The absence of a significant response in tissues such as the lung and colon suggest the possibility that these organs did not have a substantial pool of senescent cells following total body irradiation and allogeneic transplantation, as observed with the non-significant change in IL-6 mRNA in the colon and IL-6 and IL-8 mRNAs in the lung tissue (Figure 5A,B). While the source of the senescent pool is still being investigated, based on a recent report [41] and data presented in this work, it is plausible that DQ mitigates the effect of irradiation followed by allogeneic transplantation on the senescent cell pool and SASP-like factors that promote aberrant immune activation in the host tissues/organs. This would limit subsequent organ damage and other systemic effects associated with cGVHD.

## 5. Concluding Remarks

Our results support the hypothesis that accumulation of senescent cells, probably following irradiation of the recipient animal and subsequent allogeneic grafting, is a contributing factor underlying cGVHD, and its therapeutic targeting by senolytics such as DQ augurs well as a potential strategy against this life-threatening sequela of allogeneic transplantation (schematic Figure 6). Several senolytic approaches have been adopted in recent years to target senescent cells, namely, curcumin analogues, quercetin nanoparticles, exosomal carriers of embryonic stem cells, small-molecule compounds that target the anti-apoptotic protein Bcl-2, and nanoparticles targeting SASP proteins [43,44,45,46,47,48,49,50,51,52,53]. Many of these strategies have shown promise in models of cGVHD, thus supporting a role for senescent cell populations and secretory chemokines/cytokines in the pathogenesis of the disease [7,14,54,55]. More importantly, previous studies have found a profound impact of DQ in the treatment of inflammatory disorders and fibrotic disease [56,57,58]. Laying our rationale on these foundations, our results highlight the therapeutic potential of DQ in mitigating cGVHD progression through the removal of senescent cells in the liver of allograft recipients.

## Figures and Tables

**Figure 1 biology-12-00647-f001:**
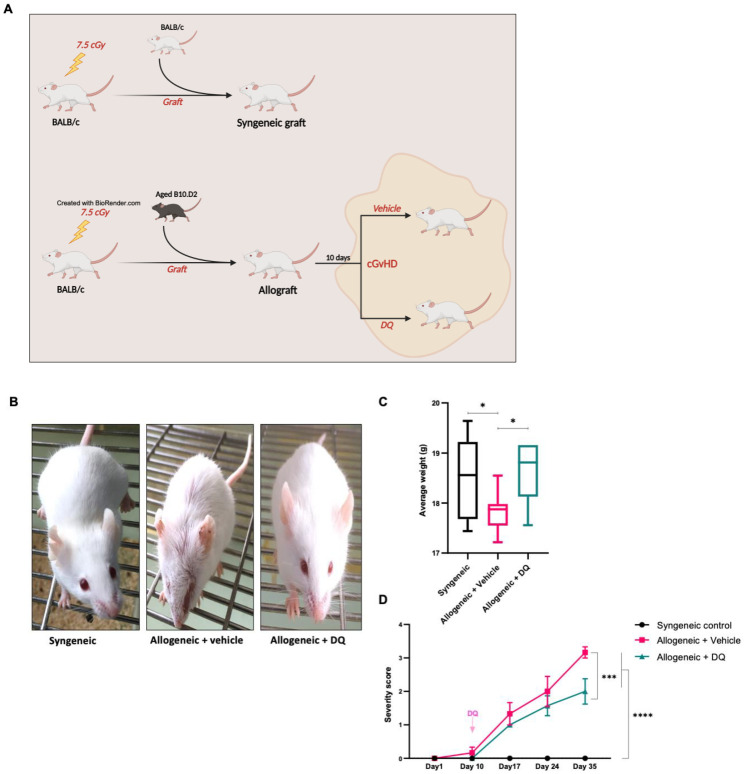
Effects of the senolytic combination of DQ on murine physical features following clinical onset of cGVHD. (**A**) Graphical representation of the study design employed. (**B**) Representative photographs of mice on day 34 post-transplantation. (**C**) Progression of the mean body weight of mice in the 3 groups over the time course of the study. One-way ANOVA was used to assess statistical significance (* = *p* < 0.05). (**D**) Mean disease severity scores (+/− SEM) over the time course of the study post-transplantation. One-way ANOVA was used to assess statistical significance (*** = *p* < 0.001, **** = *p* < 0.0001).

**Figure 2 biology-12-00647-f002:**
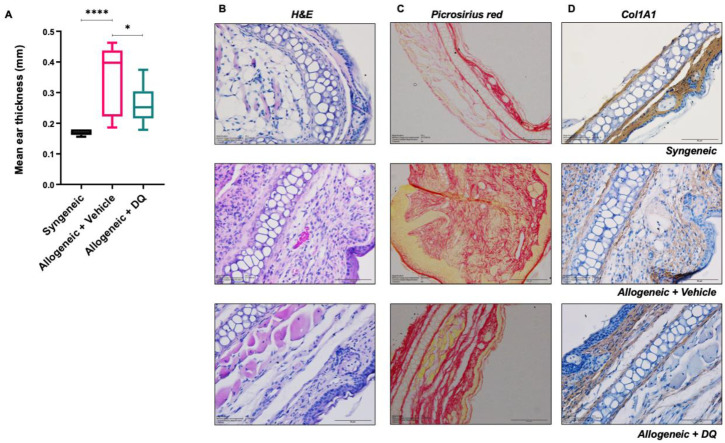
DQ treatment ameliorates cGVHD-associated fibrosis in ear and skin tissues. (**A**) Mean ear thickness was plotted over the time course of the study following grafting. One-way ANOVA was used for statistical significance (* = *p* < 0.05 and **** = *p* < 0.0001). (**B**–**D**) Representative images of H&E staining (40×) (**B**), picrosirius red stain (20×) (**C**), and col1A1 antibody (40×) (**D**) in ear tissue sections from syngeneic recipients, allogeneic recipients and allogeneic recipients treated with DQ. Allogeneic mice subjected to DQ treatment displayed less fibrosis (attributed to collagen abundance). (**E**–**I**) Representative images of H&E staining (40×) (**E**), picrosirius red stain (20×) (**F**), anti-col1A1 (40×) (**G**), MPO (40×) (**H**), and p16 (40×) (**I**) in skin tissue sections from mice in the three groups.

**Figure 3 biology-12-00647-f003:**
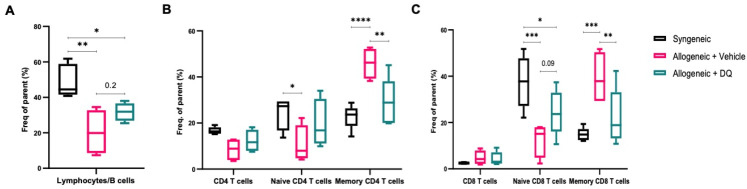
DQ treatment alleviates changes in T cell immune profile during cGVHD. (**A**) Splenocytes were subjected to FACS analysis with anti-B220-PECy7 to label CD45+ B cells amongst total lymphocyte populations. One-way ANOVA was used for statistical significance (* = *p* < 0.05, ** = *p* < 0.01). (**B**) Frequency of naïve and memory CD4 cells amongst total CD4 population was determined by CD62L and CD44 labeling. Two-way ANOVA was used for statistical significance (* = *p* < 0.05, ** = *p* < 0.01, **** = *p* < 0.0001). (**C**) Frequency of naïve and memory CD8 cells amongst total CD8 population was determined by CD62L and CD44 labeling. Two-way ANOVA was used for statistical significance (* = *p* < 0.05, ** = *p* < 0.01, *** = *p* < 0.001, **** = *p* < 0.0001).

**Figure 4 biology-12-00647-f004:**
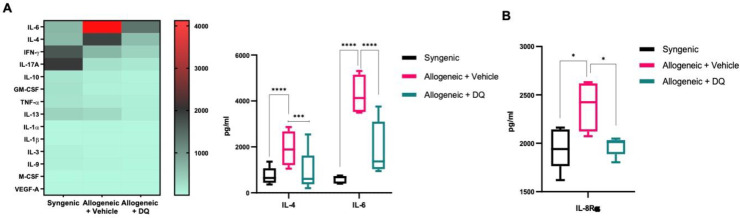
DQ treatment modulates cytokine secretion and protein expression associated with GVHD in allogeneic recipients. (**A**) 15-plex ELISA profiling of key inflammatory cytokines in the three groups of animals. IL-4 and IL-6 levels in the three groups were plotted and two-way ANOVA was used for statistical significance (*** = *p* < 0.001, **** = *p* < 0.0001). (**B**) IL-8Rα secretion in syngeneic and allograft recipients was measured using ELISA. One-way ANOVA was used for statistical significance (* = *p* < 0.05).

**Figure 5 biology-12-00647-f005:**
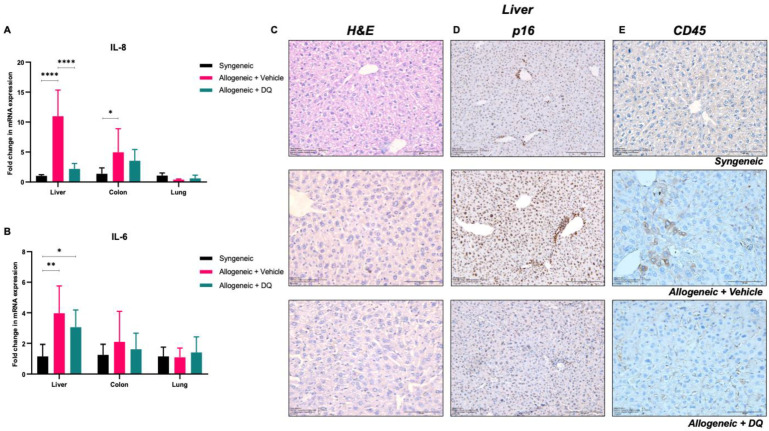
DQ targets senescent cells in liver. (**A**,**B**) mRNA expression levels of IL-8 and IL-6 produced by liver, colon, and lung cells were determined by qPCR. Two-way ANOVA was used for statistical significance (* = *p* < 0.05, ** = *p* < 0.01, **** = *p* < 0.0001). (**C**–**E**) Representative images of H&E stain (40×) (**C**), p16 (20×) (**D**), and CD45 (40×) (**E**) in liver tissue sections from mice in the syngeneic, allogeneic, and allogeneic + DQ groups.

**Figure 6 biology-12-00647-f006:**
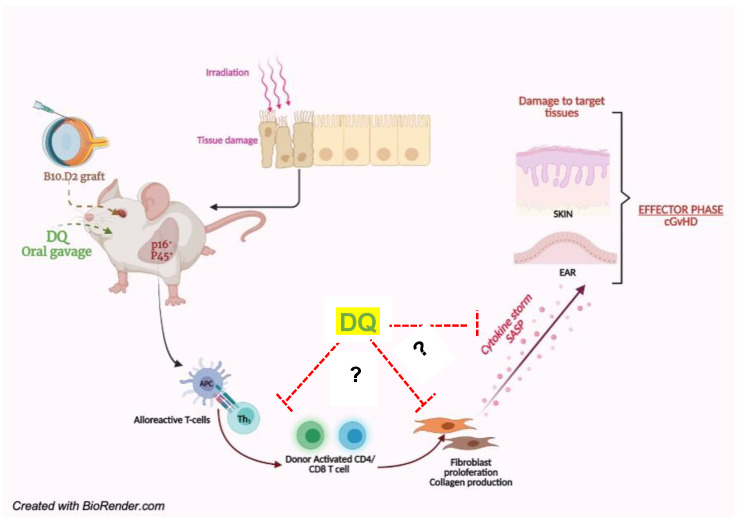
Senolytic combination of DQ alleviates organ manifestations of cGVHD, particularly skin fibrosis, accumulation of senescent cells in the liver, and alloreactive T cell repertoire. Summary model depicting the proposed mechanism of the physical and molecular pathophenotypes mitigated by targeting the senescence-associated secretory phenotype-driven cytokine storm in cGVHD. (?) denotes the probable sites of action; however, the exact mechanism of senescence inhibitory therapeutic activity of DQ remains unclear.

**Table 1 biology-12-00647-t001:** Severity score determination was performed routinely with scores rendered to determine the incidence and severity of disease. Following criteria: 0: no external sign; 1: piloerection on back and underside, 1: hunched posture or lethargy; 1: alopecia; 1: eyelid sclerosis (blepharophimosis). The severity score is the sum of these values, and ranges from 0 (unaffected) to a maximum of 4 for each mouse.

Syngeneic						Allogeneic + Vehicle						Allogeneic + DQ						
Day 10	0	0	0	0	0	0	1	0	1	0	0	0	0	0	0	0	0	0
Day 17	0	0	0	0	0	1	3	1	1	1	1	1	1	1	1	1	1	1
Day 24	0	0	0	0	0	3	3	3	1	1	1	1	3	1	2	2	1	1
Day 35	0	0	0	0	0	3	3	4	3	3	3	1	3	1	3	3	1	2

## Data Availability

All data generated during and/or analyzed during the current study are available from the corresponding author upon reasonable request.

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
