# Peer review of "Therapeutic Potential of a Senolytic Approach in a Murine Model of Chronic GVHD"

_biology, 2023, doi:10.3390/biology12050647_

Round 1

Reviewer 1 Report

Raman et al have proposed a potential role for the combination of dasatinib and quercertin [DQ] in treating chronic graft versus host disease [cGVHD] a major late cause of morbidity and morbidity after allogeneic hematopoietic cell transplantation.  In a pilot study, the authors employed an established murine model of cGVHD and investigated the role of the senolytic DQ combination in potentially targeting non-cycling senescent cells elaborating the cytokines and chemokines associated with cGVHD pathophysiology. They found that DQ administration in mismatched bone marrow allograft recipients resulted in superior general well-being to mice [by standard criteria] not receiving DQ and similar to control animals receiving syngeneic transplants. Indices of skin fibrosis showed reduced skin thickness, collagen type I and myeloperoxidase in the DQ treated versus untreated allotransplant cohort.  The studies further showed that the expected increase in CD4+ and also CD8+ memory T cells in the allotransplanted mice was reduced in the DQ treated cohort.  DQ was also shown to prevent/reduce  increases in IL-4, IL-6 and IL-8a. These observations are consistent with effects of senolytics on cells exhibiting a senescence associated secretory phenotype [SASP]. To address this the authors looked at IL-6 and IL-8 mRNA in tissues affected by cGVHD. While increased liver tissue IL-8 mRNA in allograft recipients was markedly reduced in the DQ cohort, effects in colon and lung tissue or with IL-6 for any of the RNAs of three tissues was not observed. A reduction in the senescence markers p16 and CDKN2A from DQ, however was documented in liver tissue and intestine, respectively implicating the involvement of senescence in the cGVHD model.  Perhaps the authors could speculate on the lack of response in some of the experiments involving the cGVHD targets: skin, lung and colon. The methodology employed in this pilot study is adequate but additional experiments to definitively address the hypothesis are required. For example, including an irradiated syngeneic control cohort would be helpful to better understand the role of the irradiation of transplant recipients in the development of senescent cells, as the authors suggest. They are however, likely beyond the scope of this pilot.  Overall, this is an interesting and potentially important observation that could move the field forward and perhaps help overcome one of the major and most serious challenge of allogeneic hematopoietic cell transplantation.

On a minor point, the authors should address the labeling of figures. For example, figure 2I is nowhere present in the manuscript.  

Author Response

Response:  The authors would like to thank the reviewer for finding our pilot study interesting and potentially important.  We are highly grateful to the reviewer for also recognizing that extensive additional experiments are beyond the scope of this pilot ´proof-of-concept´ study highlighting the therapeutic potential of a senolytic approach in a murine model of cGVHD. 

As for the lack of response in some of the experiments involving the target organs, we have added the following paragraph in the Discussion section:

The absence of a significant response in tissues such as the lung and colon suggest the possibility that these organs did not have a substantial pool of senescent cells following total body irradiation and allogeneic transplantation, as observed with the nonsignificant change in IL-6 mRNA in the colon and IL-6 and IL-8 mRNAs in the lung tissue (Figure 5 A & B).” 

We apologize for the oversight in the description of Figure 2I.  All captions of Figure 2 are now added in the revised version. 

Reviewer 2 Report

In this paper, Raman et al tested the effect of dasatinib and quercetin on chronic GVHD (cGVHD)after hematopoietic cell transplantation in a murine model. The authors found that the combination treatment mitigated general symptoms of cGVHD, particularly in the skin and liver. In addition, the treatment affected on the population of memory T cells and suppressed the production of pro-inflammatory cytokines, such as IL-4 and IL6. Finally, they argued the senolytic activity of dasatinib and quercetin on the cGVHD symptoms, by showing the significant reduction of the accumulation of p16 expressing cells in the liver and skin.

While I agree with the authors that there is an unmet need for identifying novel treatment strategies to alleviate cGVHD and that the treatment with dasatinib and quercetin has potential, I have several concerns on the current manuscript.

1.    The current paper is lacking the basic information why the authors have chosen the combination treatment of dasatinib and quercetin. What if each drug was administered independently? How did the authors determine the amount of 5mg/kg dasatinib and 50mg/kg quercetin?

2.    What is the origin and cell-lineage of p16+ cells in the liver and skin? 

3.    It is completely unclear what kind of mechanisms were involved in the disappearance of p16+ cells. Did the treatment of dasatinib and quercetin purge p16+ cells from the liver and skin, or only suppress the expression levels of p16 in those cells? Did the treatment induce apoptosis? Does the treatment eradicate the accumulated p16+ cells after the skin cGVHD is histologically established?

4.    Figure 1B

The authors mention that they are representative photographs of mice on day 34 post transplantation. However, the appearance of “Alloganic+DQ” mouse is very healthy, which contradict to the results summarized in Table 1. Table 1 shows the median score of cGVHD severity was “2” in the “Alloganic+DQ” group.

5.    Several figures such as Figure 1A is too fuzzy and small to be appreciated. In addition, FigureE-I referred in the text are missing, which is precluding appropriate evaluation of this paper.

Author Response

  1. The current paper is lacking the basic information why the authors have chosen the combination treatment of dasatinib and quercetin. What if each drug was administered independently? How did the authors determine the amount of 5mg/kg dasatinib and 50mg/kg quercetin?

Response: The authors would like to thank the reviewer for pointing this out.  The combination (DQ) of dasatinib, a Src/tyrosine kinase inhibitor, and quercetin, a natural flavonoid that modulates transcription factors, cell cycle proteins, pro- and anti-apoptotic proteins, growth factors and protein kinases, has been used extensively as a senolytic approach and currently undergoing clinical trials against age related pathologies. The particular dose regimen used in this study was adopted from a number of previous studies that have established and demonstrated that a combination regimen of 5mg/kg of dasatinib and 50mg/kg of quercetin elicits effective senolytic activity (targets senescent cells) in murine models (1-6). We have added the relevant citations under the Methodology section describing DQ administration.

  1. What is the origin and cell-lineage of p16+ cells in the liver and skin? 

Response: The reviewer has posed a compelling and relevant question. The association between target organ pathology in cGvHD and stress-induced cellular senescence, mediated by secretory factors released (cytokines and chemokines) upon the acquisition of SASP, has been demonstrated in several studies.  As such, the selective elimination of senescent cells with senolytic agents such as the BH3 mimetic ABT-263, results in marked improvement in target organ damage(7). Inhibiting the major components of the SASP, including IL-6 and CXCL9, with senolytics has shown promising results in sclerodermatous cGvHD mouse model(7). In our present pilot study, we used the same sclerodermatous murine model of cGvHD and investigated the therapeutic potential of a drug combination (DQ) that has been shown to eliminate senescent cells by inducing senolysis.  We observed an increase in SASP factors such as IL-6, IL-8Ra as well as p16+ cell population in the liver and skin of allogeneic transplant recipient mice.  P16 is a marker of cellular senescence and induced upon DNA damage.  Notably, DQ treated animals showed a significantly reduced levels of the SASP cytokines as well as p16 positivity, particularly in the liver and skin. While the source of the p16+ cells remains to be determined, we believe that pre-conditioning of the host with total body irradiation results in radiation-induced DNA damage and irreversible cell cycle arrest (senescence).  This is evidenced by the increase in p16+ cells in allo-recipients, which then fuels the secretion of pro-inflammatory SASP factors.  Notably, treatment with DQ resulted in a significantly lower p16+ cells thus indicating the senescence targeting effect of DQ.  

  1. It is completely unclear what kind of mechanisms were involved in the disappearance of p16+ cells. Did the treatment of dasatinib and quercetin purge p16+ cells from the liver and skin, or only suppress the expression levels of p16 in those cells? Did the treatment induce apoptosis? Does the treatment eradicate the accumulated p16+ cells after the skin cGVHD is histologically established?

Response:  In this pilot study, cGvHD was first induced in the irradiated recipient mice by allogeneic transplantation from MHC mismatched donors.  Importantly, the selection of DQ was based on its reported ability to target senescent cell pool (senolytic strategy), as we hypothesized that senescence associated cytokines and chemokines drive tissue and organ damage in cGvHD.  As DQ efficiently targets senescent cells, the marked reduction in p16, a marker of senescence, seems to suggest purging of p16+ cells; however, this warrants further investigation and may be determined by using a p16+ or SA-β-Gal+ reporter (8, 9) model to track senescent cell population(s) in cGvHD and the effect of DQ. As mentioned earlier, the skin cGvHD is first established before treatment with DQ, which therefore suggests that DQ reduces the number of p16+ cells via its senolytic activity.  We have included this explanation in the Results section of the manuscript.  We would like to point out to the reviewer that the purpose of this pilot study is to establish a proof-of-concept model to test the therapeutic potential of senolysis using DQ as a senolytic in a murine model of cGvHD.  While we agree with the reviewer on the need to perform follow up study addressing the underlying mechanisms(s) in a more detailed manner, we sincerely hope that the reviewer will appreciate the overall message of this pilot study and find it acceptable for publication.   

  1. Figure 1B: The authors mention that they are representative photographs of mice on day 34 post transplantation. However, the appearance of “Alloganic+DQ” mouse is very healthy, which contradict to the results summarized in Table 1. Table 1 shows the median score of cGVHD severity was “2” in the “Alloganic+DQ” group.

Response: The severity score is calculated based on multiple parameters that may not be apparent phenotypically.  For example, a score of 1 was automatically assigned to all mice in the group (in the cage) with diarrhea. This for example could not be assessed as an individual character and thus increases the overall score.  What we see in the DQ treated mouse is the obvious improvement in cGvHD associated skin lesions and ear fibrosis, which is corroborated by the data on the histology of the skin and liver. On the other hand, the disease severity score seems to indicate that not all parameters associated with cGvHD might be affected upon DQ treatment. 

  1. Several figures such as Figure 1A is too fuzzy and small to be appreciated. In addition, FigureE-I referred in the text are missing, which is precluding appropriate evaluation of this paper.

Response: We apologize for the oversight.  We have included improved quality of Figure 1.  Figures 2E-I were somehow removed from our submission but have been included in the revised version. 

 References

  1. Saccon TD, Nagpal R, Yadav H, Cavalcante MB, Nunes ADC, Schneider A, et al. Senolytic Combination of Dasatinib and Quercetin Alleviates Intestinal Senescence and Inflammation and Modulates the Gut Microbiome in Aged Mice. J Gerontol A Biol Sci Med Sci. 2021;76(11):1895-905.
  2. Ota H, Kodama A. Dasatinib plus quercetin attenuates some frailty characteristics in SAMP10 mice. Scientific Reports. 2022;12(1):2425.
  3. Novais EJ, Tran VA, Johnston SN, Darris KR, Roupas AJ, Sessions GA, et al. Long-term treatment with senolytic drugs Dasatinib and Quercetin ameliorates age-dependent intervertebral disc degeneration in mice. Nature Communications. 2021;12(1):5213.
  4. Ermogenous C, Green C, Jackson T, Ferguson M, Lord JM. Treating age-related multimorbidity: the drug discovery challenge. Drug discovery today. 2020;25(8):1403-15.
  5. Hickson LJ, Langhi Prata LGP, Bobart SA, Evans TK, Giorgadze N, Hashmi SK, et al. Senolytics decrease senescent cells in humans: Preliminary report from a clinical trial of Dasatinib plus Quercetin in individuals with diabetic kidney disease. EBioMedicine. 2019;47:446-56.
  6. Cavalcante MB, Saccon TD, Nunes ADC, Kirkland JL, Tchkonia T, Schneider A, et al. Dasatinib plus quercetin prevents uterine age-related dysfunction and fibrosis in mice. Aging (Albany NY). 2020;12(3):2711-22.
  7. Yamane M, Sato S, Shimizu E, Shibata S, Hayano M, Yaguchi T, et al. Senescence-associated secretory phenotype promotes chronic ocular graft-vs-host disease in mice and humans. Faseb j. 2020;34(8):10778-800.
  8. Wang Y, Liu J, Ma X, Cui C, Deenik PR, Henderson PKP, et al. Real-time imaging of senescence in tumors with DNA damage. Sci Rep. 2019;9(1):2102.
  9. Omori S, Wang TW, Johmura Y, Kanai T, Nakano Y, Kido T, et al. Generation of a p16 Reporter Mouse and Its Use to Characterize and Target p16(high) Cells In Vivo. Cell metabolism. 2020;32(5):814-28.e6.

Round 2

Reviewer 2 Report

In this revised manuscript, the authors have provided background rationale of dasatinib and quercetin treatment by citing related papers. This reviewer now understands the authors’ claim that the main purpose of this study is to establish a proof-of-concept model to test the therapeutic potential of senolysis using dasatinib and quercetin on murine cGvHD. Although the underlying mechanisms remain to be elucidated, this reviewer agrees that those issues are beyond the scope of this pilot study.